# Consumption of Ultra-Processed Food and Drinks and Chronic Lymphocytic Leukemia in the MCC-Spain Study

**DOI:** 10.3390/ijerph18105457

**Published:** 2021-05-20

**Authors:** Marta Solans, Sílvia Fernández-Barrés, Dora Romaguera, Yolanda Benavente, Rafael Marcos-Gragera, Esther Gracia-Lavedan, Laura Costas, Claudia Robles, Eva Gonzalez-Barca, Esmeralda de la Banda, Esther Alonso, Marta Aymerich, Elias Campo, Javier Llorca, Guillermo Fernández-Tardón, Rocío Olmedo-Requena, Eva Gimeno, Gemma Castaño-Vinyals, Nuria Aragonés, Manolis Kogevinas, Marina Pollán, Silvia de Sanjose, Pilar Amiano, Delphine Casabonne

**Affiliations:** 1Centro de Investigación Biomédica en Red: Epidemiología y Salud Pública (CIBERESP), 28001 Madrid, Spain; marta.solans@udg.edu (M.S.); silvia.fernandez@isglobal.org (S.F.-B.); ybenavente@iconcologia.net (Y.B.); rafael.marcos@udg.edu (R.M.-G.); esthergrala@gmail.com (E.G.-L.); javier.llorca@unican.es (J.L.); gfernanta@gmail.com (G.F.-T.); rocioolmedo@ugr.es (R.O.-R.); gemma.castano@isglobal.org (G.C.-V.); nuria.aragones@salud.madrid.org (N.A.); manolis.kogevinas@isglobal.org (M.K.); mpollan@isciii.es (M.P.); sdesanjose@path.org (S.d.S.); epicss-san@euskadi.eus (P.A.); 2Research Group on Statistics, Econometrics and Health (GRECS), University of Girona, 17003 Girona, Spain; 3ISGlobal, 08036 Barcelona, Spain; dora.romaguera@isglobal.org; 4Departament de Ciències Experimentals i de la Salut, Universitat Pompeu Fabra (UPF), 08002 Barcelona, Spain; 5Health Research Institute of the Balearic Islands (IdISBa), University Hospital Son Espases, 07120 Palma de Mallorca, Spain; 6CIBER Fisiopatología de la Obesidad y Nutrición (CIBEROBN), Instituto de Salud Carlos III (ISCIII), 28001 Madrid, Spain; 7Unit of Molecular and Genetic Epidemiology in Infections and Cancer (UNIC-Molecular), Cancer Epidemiology Research Programme, IDIBELL, Catalan Institute of Oncology, 08908 L’Hospitalet de Llobregat, Spain; lcostas@iconcologia.net; 8Epidemiology Unit and Girona Cancer Registry, Catalan Institute of Oncology, 17007 Girona, Spain; 9Josep Carreras Leukemia Research Institute, 08916 Badalona, Spain; 10Unit of Information and Interventions in Infections and Cancer (UNIC-I&I), Cancer Epidemiology Research Programme, (IDIBELL), Catalan Institute of Oncology, 08908 L’Hospitalet de Llobregat, Spain; crobles@idibell.cat; 11Hematology, Bellvitge Biomedical Research Institute (IDIBELL), Catalan Institute of Oncology, 08908 L’Hospitalet de Llobregat, Spain; e.gonzalez@iconcologia.net; 12Hematology Laboratory, Department of Pathology, Hospital Universitari de Bellvitge, 08908 L’Hospitalet de Llobregat, Spain; edelabanda@bellvitgehospital.cat (E.d.l.B.); ealonso@bellvitgehospital.cat (E.A.); 13Hematopathology Section, Hospital Clinic de Barcelona, University of Barcelona, CIBERONC, 08001 Barcelona, Spain; AYMERICH@clinic.ub.es (M.A.); ECAMPO@clinic.cat (E.C.); 14Faculty of Medicine, University of Cantabria—IDIVAL, 39011 Santander, Spain; 15Instituto de Investigación Sanitaria del Principado de Asturias (ISPA), Universidad de Oviedo, 33003 Oviedo, Spain; 16Department of Preventive Medicine and Public Health, University of Granada, 18071 Granada, Spain; 17Instituto de Investigación Biosanitaria (ibs.GRANADA), 18012 Granada, Spain; 18Hematology Department, Hospital del Mar, 08003 Barcelona, Spain; 94015@parcdesalutmar.cat; 19Hospital del Mar Medical Research Institute (IMIM), 08001 Barcelona, Spain; 20Epidemiology Section, Public Health Division, Department of Health, 28009 Madrid, Spain; 21Cancer Epidemiology Unit, National Centre for Epidemiology, Instituto de Salud Carlos III, 28029 Madrid, Spain; 22Sexual and Reproductive Health, PATH, Seattle, WA 98121, USA; 23Ministry of Health of the Basque Government, Sub-Directorate for Public Health and Addictions of Gipuzkoa, 20013 San Sebastián, Spain; 24Biodonostia Health Research Institute, Group of Epidemiology of Chronic and Communicable Diseases, 20014 San Sebastián, Spain

**Keywords:** chronic lymphocytic leukemia, cancer, NOVA classification, ultra-processed food, case-control study

## Abstract

Chronic lymphocytic leukemia (CLL) is the most common leukemia in adults in Western countries. Its etiology is largely unknown but increasing incidence rates observed worldwide suggest that lifestyle and environmental factors such as diet might play a role in the development of CLL. Hence, we hypothesized that the consumption of ultra-processed food and drinks (UPF) might be associated with CLL. Data from a Spanish population-based case-control study (MCC-Spain study) including 230 CLL cases (recruited within three years of diagnosis) and 1634 population-based controls were used. The usual diet during the previous year was collected through a validated food frequency questionnaire and food and drink consumption was categorized using the NOVA classification scheme. Logistic regression models adjusted for potential confounders were used. Overall, no association was reported between the consumption of UPF and CLL cases (OR per each 10% increase of the relative contribution of UPF to total dietary intake = 1.09 (95% CI: 0.94; 1.25)), independently of the Rai stage at diagnosis. However, when analyses were restricted to cases diagnosed within <1 year (incident), each 10% increment in the consumption of UPF was associated with a 22% higher odds ratio of CLL (95% CI: 1.02, 1.47) suggesting that the overall results might be affected by the inclusion of prevalent cases, who might have changed their dietary habits after cancer diagnosis. Given the low number of cases in the subgroup analyses and multiple tests performed, chance findings cannot totally be ruled out. Nonetheless, positive associations found in CLL incident cases merit further research, ideally in well-powered studies with a prospective design.

## 1. Introduction

Chronic lymphocytic leukemia (CLL), a subtype of non-Hodgkin lymphoma (NHL), is the most common leukemia in adults in Western countries with an incidence rate of 5 per 100,000 person-years in Europe [1]. The disease, characterized by an abnormal accumulation of B-lymphocytes in blood, bone marrow and/or lymphoid tissues, typically occurs in the elderly and shows a highly heterogeneous clinical course. While the majority of patients live for decades without being treated and die from other causes, others show a rapidly aggressive course and premature death [2]. In line with other hematological neoplasms, the etiology of CLL is largely unknown with a few well-established risk factors including increasing age, male sex, being Caucasian and a familial history of hematological neoplasms and several suggestive associations yet to be confirmed [3]. However, the high incidence of CLL in Western regions [4], the increasing CLL incidence rates across the world [5] and the change in CLL incidence patterns with acculturation as seen in higher incidence rates among Asian US born than Asian foreign born [6,7] suggest that lifestyle and environmental factors may also play an important role CLL etiology.

With regard to dietary exposures, the 2007 report by the World Cancer Research Fund/American Institute for Cancer Research (WCRF/AICR) [8] pointed out a suggestive inverse association between NHL and the consumption of vegetables, fruit and alcoholic beverages and a positive association with meat, total fat, body fatness and dairy. Recent meta-analyses further support these associations [9,10,11,12,13] but data are still inconclusive, particularly for specific NHL subtypes. Specifically, in CLL, evidence regarding dietary exposures and the risk of CLL, which mainly arises from studies assessing single food items [14], is inconsistent. However, the few studies that have examined dietary patterns have reported an association between an adherence to a Western-like diet with NHL [15] and CLL [14]. Whether the presence of ultra-processed food and drinks (UPF) in a western dietary pattern (typically high in saturated fats, processed meat, refined grains, sweets, caloric drinks and convenience foods) is contributing to the associations found is not known.

According to the NOVA classification [16], UPF are industrial formulations typically with five of more ingredients made mostly or entirely from substances derived from foods and additives. Following multiple biological, physical and chemical processes, these ultimate products are conceived to be safe, highly palatable and affordable. However, other less desirable properties [17,18] including their poor nutritional composition (i.e., high amounts of saturated fats, refined starches, free sugars and salts), the potential presence of components derived from food processing (i.e., heterocycle amines, acrylamide or aromatic polycyclic hydrocarbons) or packaging (i.e., bisphenol A) might have a deleterious effect on human health [17]. Indeed, cumulative evidence from epidemiological studies points to an association between the consumption of UPF and obesity [19], cardiovascular disease [20], mortality [21,22,23] and, more recently, cancer [24,25]. Regarding the latter, a French prospective study conducted in 104,980 participants reported an increased risk of developing cancer and, more specifically, breast cancer [24]. Lately, in a large Spanish multicase-control study (MCC-Spain) including cases of colorectal (*n* = 1852), breast (*n* = 1486) and prostate cancer (*n* = 953), the consumption of UPF was associated with colorectal cancer [25]. However, to date, no study has assessed their potential impact on hematological neoplasms.

Considering this, the aim of the present study was to evaluate whether the consumption of UPF was associated with CLL in the MCC-Spain study.

## 2. Materials and Methods

### 2.1. Study Population

MCC-Spain (http://www.mccspain.org) (accessed on 11 January 2021) was a population-based multicenter case-control study conducted in Spain in 2008–2013 [26]. Adult CLL cases were recruited in eleven hospitals from five Spanish provinces (i.e., Asturias, Barcelona, Cantabria, Girona and Granada). The controls were randomly selected from primary health centers and were frequency matched by sex, geographical area and age. The participation rates were 87% in cases and 53% in controls, which varied by region. After excluding subjects with no dietary data, implausible energy intakes <750 or >4500 kcal/day and CLL cases recruited more than three years after diagnosis, a total of 1634 controls and 230 CLL cases were included (Appendix A) in the present study.

### 2.2. Outcome Definition

Both cases of CLL and small lymphocytic lymphoma (SLL) were recruited, as they are considered to be the same underlying disease [27]. All diagnoses were histologically and morphologically confirmed and fulfilled the 2008 WHO criteria [27]. CLL/SLL cases were identified from March 2010 to November 2013 in hospitals from five Spanish regions (i.e., Barcelona, Girona, Granada, Asturias and Cantabria). Given the indolent course of the disease, newly diagnosed cases as well as patients with a CLL/SLL diagnosis prior to the start of the study (up to three years) were included. Thus, separate analyses for incident cases (time from diagnosis to interview <1 year, *n* = 97) and prevalent cases (time from diagnosis to interview 1–3 years, *n* = 133) were performed as sensitivity analyses. In addition, the disease severity was assessed using the Rai staging system obtained from medical records and verified by local hematologists. The cases were then categorized into a low-risk category (Rai 0, *n* = 138) and an intermediate/high-risk category (Rai I–IV, *n* = 81).

### 2.3. Data Collection

Trained personnel collected data on socio-demographic factors, lifestyle and personal/family medical history through face-to-face interviews using a computerized questionnaire. Dietary data were collected using a 140 item semi-quantitative food frequency questionnaire (FFQ), which assessed the usual food intake during the previous year. The FFQ was an adapted version of a Spanish validated FFQ, modified to include several Spanish regional foods [28]. The FFQ was self-administered and returned by mail or filled out in face-to-face interviews. The nutrient intake was calculated using Spanish food composition [29]. Cross-check questions on aggregated food group intakes were used to reduce the misreporting of food groups with a large number of items and to adjust the frequency of food consumption [30].

### 2.4. Exposure: UPF

Food and drink items of the FFQ were categorized into one of the four groups in NOVA, a food classification system based on the extent and purpose of industrial food processing [25]. For several items, not enough information was available in the FFQ to unequivocally classify them into one of the four NOVA categories. In such cases, a consensus was reached based on the food composition table used [31], the labels of the products and information of the usual consumption from household consumption databases [32]. The detailed description of these variables and the assumptions made are provided in the Appendix A. This study primarily focused on group 4 of the NOVA classification, which includes industrial formulations that have usually added industrial ingredients such as hydrogenated oils, flavors or emulsifiers and contain little or no whole foods (e.g., sweet or savory packaged snacks, sweetened beverages and ready to eat foods). Following previous studies [24,25], for each individual the consumption of all items included in each NOVA group (expressed in daily grams) was added and expressed as a percentage of the total dietary intake (daily g within the NOVA category/total daily g, multiplied by 100). This variable was further categorized into low, medium and high consumption according to sex-specific tertiles in the control group.

### 2.5. Statistical Analyses

We performed descriptive analyses of anthropometric, socio-demographic and lifestyle characteristics. The differences between cases and controls and across tertiles of the consumption of UPF were assessed using a Student’s *t*-test (or ANOVA test, when appropriate) and a Pearson chi-squared test.

We used multivariable logistic regression models to evaluate the association between UPF and CLL. Odds ratios (OR) and 95% confidence intervals (CI) were calculated. The UPF were analyzed as a continuous variable (per 10% increment) and as a categorical variable (low, medium, high consumption). The first tertile (low consumption) was considered to be a reference category. P-value for the trend was calculated including the categorical variable as a continuous ordinal (scored from 1 to 3) in our models.

Two models with two levels of adjustment were used. The basic model (model 1) included as covariates age, sex, educational level and province. The multiple adjusted model (model 2) further included a family history of hematologic malignancies, having ever worked in farming or agriculture, energy intake, body mass index, ethanol intake, physical activity in the last ten years up to two years before diagnosis and smoking status. Missing values in the categorical variables were coded as a separate category (Table 1). Model 2 was also run after stratification according to a series of variables that might influence the association between UPF and CLL (i.e., age, sex, smoking and alcohol consumption). The effect modifications were explored by modelling interaction terms between those variables and the (continuous) consumption of UPF and tested using log-likelihood ratio tests.

In line with previous studies on UPF [24,25], we performed complementary analyses by further adjusting model 2 for dietary factors that could act as potential confounders of the association: energy density (kcal), fiber intake (g/day), sugar intake (% of total energy intake), saturated fatty acid intake (% of total energy intake) and fruit and vegetable consumption (g/day).

As we assessed the usual diet during the previous year, in cases interviewed within ≥1 year from diagnosis, we captured a post-diagnosis diet. To ensure that potential dietary changes or survival bias might not be influencing our results, we performed stratified analyses by time from diagnosis to interview (<1 year, i.e., incident cases and ≥1 year, i.e., prevalent cases). Likewise, we performed stratified analyses by the severity of disease (Rai 0 vs. Rai 1–4). Finally, sensitivity analyses excluding cases treated before the interview (*n* = 34) were performed.

All analyses were performed using free software R (R version 3.5.2) [33] and the statistical significance was set at two-sided *p* < 0.05.

## 3. Results

The distribution of baseline characteristics between cases and controls is shown in Table 1. Compared with the controls, the cases were slightly older, had a lower alcohol intake and were more likely to have a familial history of hematological neoplasms and to have ever worked in farming/agriculture. In the controls, on average, the relative contribution of each NOVA group to the total food intake was: group 4: 13.8% (SD: 10.6), group 3: 17.5% (SD: 11.0), group 2: 1.6% (SD: 1.1) and group 1: 67.0% (SD: 13.8). Beverages, sugary products, ready to eat products and processed meats were the UPF most frequently consumed among group 4 (Appendix A).

The distribution of control characteristics according to the level of the consumption of UPF is displayed in Table 2. Those participants in tertile 3 of the consumption of UPF (relative contribution of UPF to total food intake = 25.7% (SD 9.6%)) were on average younger, with a higher energy intake, smokers, less physically active and less likely to have worked in agriculture (all *p*-values < 0.05). No differences across the different levels of exposure were observed for other lifestyle factors. The same analyses, restricted to cases, are provided in the Appendix A.

The consumption of UPF was not associated with overall CLL both when modelled as a categorical (OR_high.vs.low_ 1.09 (95% CI 0.74; 1.60), *p*-trend = 0.665) or as a continuous variable (OR per each 10% increment in the consumption of UPF = 1.09 (95% CI: 0.94; 1.25)) (Table 3). The findings were similar regardless of the disease severity (Rai stage) (Table 3) and did not materially change when excluding cases treated before the interview (*n* = 34, data not shown). However, the results differed markedly between the incident and prevalent cases (Table 3). In the incident cases, each 10% increment in the consumption of UPF was associated with a 22% higher odds ratio of having CLL (95% CI: 1.02, 1.47). By contrast, null associations were reported in prevalent cases.

Figure 1 shows the association between UPF (per 10% increment) and CLL using model 2 further adjusted for several nutritional variables that could act as confounders or mediators of the association. The association between the consumption of UPF and CLL in incident cases remained when dietary fiber, saturated fat, simple carbohydrates, energy density and fruit and vegetables were included in the adjustment while the results for all CLLs continued to be null.

In the Appendix A, the effect modification and stratified analyses by age, sex and lifestyle factors are shown. All *p*-values for interaction were non-statistically significant, indicative of no effect-measure modification. In stratified analyses, the consumption of UPF was significantly associated with CLL (both in total and incident cases) in women and non-smokers.

## 4. Discussion

In this study, we did not find a clear pattern of association between UPF and CLL overall and by Rai stage at diagnosis. However, in analyses restricted to incident cases, each 10% increment in the consumption of UPF was associated with a 22% higher odds ratio of CLL, suggesting that such exposure might be associated with CLL.

Since the development of the NOVA classification [16], numerous observational studies have associated the consumption of UPF with mortality [21,22,23] and several non-communicable diseases such as obesity [19] and cardiovascular disease [20]. In 2018, the results from the French NutriNet-Santé prospective cohort pointed, for the first time, to a link between the consumption of such items and cancer, particularly overall and breast cancer but not for colorectal or prostate cancers [24]. Such analyses were latterly replicated in the MCC-Spain study, which revealed an association with colorectal cancer (OR_T1vs.T3_ = 1.30, 95% CI 1.11, 1.51) and null associations for breast or prostate cancers [25]. Our results, conducted in the same study, did not conclusively support an association for CLL. However, we noted a suggestive association in incident cases (*n* = 94, OR: 1.22, 95% CI: 1.02, 1.47), in contrast to those with a longer time from diagnosis to interview (*n* = 128, OR: 0.95, 95% CI: 0.78, 1.16). Although CLL is typically an indolent disease, we cannot totally rule out that the inclusion of prevalent cases might have attenuated our estimates due to survival bias or, furthermore, dietary changes after diagnosis. Thus, the results in incident cases point to a potential association between the consumption of UPF and CLL, which should be clarified in further studies ideally with a prospective design and more statistical power.

Several hypotheses have been suggested to explain the association between the consumption of UPF and cancer risk. One relates to the poorer nutritional quality of diets rich in processed foods, which tend to be rich in sugars, fats, salt and excessive energy [17,34]. In our study, however, the association of UPF with CLL in incident cases remained statistically significant after the adjustment for these dietary factors, suggesting, in line with other studies [24], the involvement of other components (e.g., additives or bioactive compounds derived from processing/packaging) beyond “nutritional quality”. Overall, mediation studies are needed to better understand the relative role of nutritional composition, additives and neoformed contaminants in this relationship.

Traditionally, Spain has been a country with a relatively low energy contribution of UPF in comparison with other Western regions such as the United States (2007–2012, 58.5%) [35], UK (2008–2012, 53%) [36] and France (2009–2014, 35.9%) [37]. According to previous data from the Data Food Networking Database repository (2000) [38] or a national prospective cohort (2008–2010) [21], UPF account for 20–25% of the total dietary calories in Spain. Thus, in our study, we dealt with an overall lower exposure to UPF with less contrast between the extreme categories, which may have attenuated our risk estimates. Thus, further analyses in forthcoming studies including younger subjects or settled in countries with greater levels of the consumption of UPF are warranted to confirm our findings.

Several limitations must be acknowledged when interpreting our results. Firstly, case-control studies are prone to be affected by recall and selection biases. Measurement errors in the estimation of dietary intake are also likely although we used a previously validated FFQ for Spanish populations. In addition, several food items could not be unequivocally classified into one of the four NOVA groups, given that our FFQ collected limited information regarding food processing (e.g., place of meals and product brand) (see Appendix A). In such cases, we used information on food composition in Spain to classify such food items and, thus, a degree of misclassification is expectable. The parameters used to construct our consumption score for UPF were based on the food intake during the previous year to recruitment. We assumed that the measured dietary exposure at the baseline actually reflected the eating habits of subjects during adulthood. However, if dietary changes had occurred during that period, we might have modelled a “recent exposure” that might not be associated with CLL lymphomagenesis in which risk factors are expected to take decades to contribute to disease development. This is particularly relevant for prevalent cases in case they changed their habitual diet after cancer diagnosis. We may have been limited by a lack of statistical power to detect significant associations when running subgroup analyses. In addition, these associations should be interpreted in the context of multiple comparisons and the possibility of chance findings. Finally, although we adjusted for a range of potential confounders, residual confounding cannot be totally ruled out. In particular, we lacked the data of several conditions such as autoimmune diseases and allergies that have been consistently associated with CLL [3] and that should ideally be considered in future studies.

## 5. Conclusions

Findings from this Spanish population-based case-control study did not conclusively support an association between the consumption of UPF and CLL; however, the results may be biased due to the inclusion of prevalent cases. The association reported in the analyses restricted to incident cases pointed to a potential role of UPF in CLL. Further studies, ideally with a prospective design and more statistical power, are required to confirm these findings.

## Figures and Tables

**Figure 1 ijerph-18-05457-f001:**
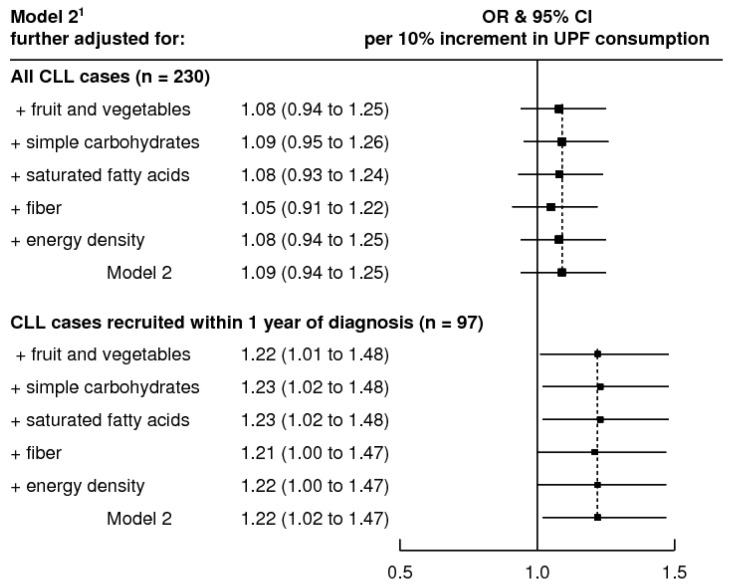
Association between a 10% increment in the consumption of ultra-processed food and drink and chronic lymphocytic leukemia in the MCC-Spain study, further adjusted for nutritional characteristics for all cases (top graph) and restricted to cases recruited within one year of diagnosis (incident; bottom graph). OR, odds ratio; 95% CI, 95% confidence interval; CLL, chronic lymphocytic leukemia; UPF, ultra-processed food and drinks. ^1^ Logistic regression adjusted for age, sex, province, educational level, family history of hematological neoplasms, ever worked in farming, physical activity, energy intake, ethanol intake and smoking status. Black squares and horizontal lines indicate the OR and 95% CI, respectively.

**Table 1 ijerph-18-05457-t001:** Baseline characteristics of controls and cases in the MCC-Spain study.

	Controls (*n* = 1634)	Cases (*n* = 230)	*p*-Value ^2^
Consumption of UPF (g/day), mean (SD)	277.3 (280.8)	298.9 (335.7)	0.353
Consumption of UPF (%) ^1^, mean (SD)	13.8 (10.6)	14.4 (11.6)	0.466
Province, *n* (%)			**<0.001**
Barcelona	897 (54.9)	148 (64.3)	
Asturias	211 (12.9)	29 (12.6)	
Cantabria	308 (18.8)	7 (3)	
Granada	146 (8.9)	22 (9.6)	
Girona	72 (4.4)	24 (10.4)	
Age (years), mean (SD)	63.9 (10.8)	65.7 (10.1)	**0.01**
Sex, *n* (%)			0.07
Male	944 (57.8)	148 (64.3)	
Female	690 (42.2)	82 (35.7)	
Education, *n* (%)			0.718
Primary	871 (53.3)	129 (56)	
Secondary	476 (29.1)	62 (27)	
University	287 (17.6)	39 (17)	
Body mass index ^3^ (kg/m^2^), mean (SD)	26.9 (4.4)	27.3 (4.2)	0.195
Energy intake (kcal/day), mean (SD)	1902.5 (585.9)	1972.4 (615.7)	0.106
Alcohol consumption, median (IQR)	5.6 (0.8–16.5)	5.1 (0.4–11.6)	**0.007**
Smoking status, *n* (%)			0.468
Never	711 (43.5)	93 (40.4)	
Former	628 (38.4)	98 (42.6)	
Current	290 (17.7)	37 (16.1)	
Unknown	5 (0.3)	2 (0.9)	
Physical activity ^4^, *n* (%)			0.720
Inactive	669 (40.9)	87 (37.8)	
Moderately active	224 (13.7)	32 (13.9)	
Active	194 (11.9)	29 (12.6)	
Very active	510 (31.2)	80 (34.8)	
Unknown	37 (2.3)	2 (0.9)	
Ever worked in farming or agriculture, *n* (%)			**<0.001**
No	1301 (79.6)	151 (65.7)	
Yes	330 (20.2)	78 (33.9)	
Unknown	3 (0.2)	1 (0.4)	
Family history of hematological malignancy, *n* (%)			
No	1445 (88.4)	193 (83.9)	**<0.001**
Yes	81 (5)	26 (11.3)	
Unknown	108 (6.6)	11 (4.8)	
Rai stage, *n* (%)			
0	-	138 (60.0)	
1–4	-	88 (38.3)	
Unknown	-	4 (1.7)	

UPF, ultra-processed food and drinks (based on group 4 of the NOVA classification); SD, standard deviation; IQR, interquartile range. ^1^ Calculated as daily g of UPF/total daily g, multiplied by 100. ^2^
*p*-value for heterogeneity calculated with the Student’s *t*-test for continuous variables and with the chi-squared test for categorical variables. Missing values were excluded from these tests. ^3^ There were 76 individuals (8 cases, 68 controls) with missing information on body mass index. ^4^ Physical activity in the last 10 years up to 2 years before diagnosis measured in METs/week: inactive (0), low (0.1–8), moderate (8–15.9) and very active (≥16). In bold: *p*-value < 0.05.

**Table 2 ijerph-18-05457-t002:** Characteristics of participants in the control group according to the consumption of ultra-processed food (NOVA classification, group 4) in the MCC-Spain study.

	Consumption of Ultra-Processed Food and Drinks ^1^			
	Low (*n* = 546)	Medium (*n* = 546)	High (*n* = 542)	*p*-Value ^3^
Consumption of UPF (%) ^2^, mean (SD)	4.5 (1.9)	11.3 (2.3)	25.7 (9.6)	**<0.001**
In men, mean (min–max)	4.8 (0.0–7.7)	11.3 (7.7–15.6)	25.3 (15.7–69.24)	
In women, mean (min–max)	4.2 (0.0–7.5)	11.4 (7.5–16.0)	26.4 (16.1–66.1)	
Province, *n* (%)				**0.018**
Barcelona	289 (52.9)	300 (54.9)	308 (56.8)	
Asturias	85 (15.6)	71 (13)	55 (10.1)	
Cantabria	92 (16.8)	107 (19.6)	109 (20.1)	
Granada	63 (11.5)	39 (7.1)	44 (8.1)	
Girona	17 (3.1)	29 (5.3)	26 (4.8)	
Age (years), mean (SD)	67.2 (9.1)	63.6 (10.7)	60.9 (11.6)	**<0.001**
Sex, *n* (%)				0.926
Male	316 (57.9)	312 (57.1)	316 (58.3)	
Female	230 (42.1)	234 (42.9)	226 (41.7)	
Education, *n* (%)				0.171
Primary	310 (56.8)	294 (53.8)	267 (49.3)	
Secondary	149 (27.3)	155 (28.4)	172 (31.7)	
University	87 (15.9)	97 (17.8)	103 (19)	
Body mass index ^4^ (kg/m^2^), mean (SD)	26.9 (4.2)	26.8 (4.4)	27.0 (4.5)	0.884
Energy intake (kcal/day), mean (SD)	1723.3 (495.0)	1925.8 (543.3)	2059.5 (659.0)	**<0.001**
Alcohol consumption, median (IQR)	6.4 (0.6–16.9)	5.1 (0.6–18.6)	5.2 (0.8–15.0)	0.214
Smoking status, *n* (%)				**<0.001**
Never	268 (49.1)	222 (40.7)	221 (40.8)	
Former	215 (39.4)	216 (39.6)	197 (36.3)	
Current	61 (11.2)	108 (19.8)	121 (22.3)	
Unknown	2 (0.4)	0	0	
Physical activity ^5^, *n* (%)				**0.003**
Inactive	211 (38.6)	211 (38.6)	247 (45.6)	
Moderately active	63 (11.5)	86 (15.8)	75 (13.8)	
Active	81 (14.8)	53 (9.7)	60 (11.1)	
Very active	184 (33.7)	182 (33.3)	144 (26.6)	
Unknown	7 (1.3)	14 (2.6)	16 (3)	
Ever worked in farming or agriculture, *n* (%)				**<0.001**
No	404 (74)	439 (80.4)	458 (84.5)	
Yes	142 (26)	107 (19.6)	81 (14.9)	
Unknown	0	0	3 (0.6)	
Family history of hematological malignancy, *n* (%)				0.948
No	483 (88.5)	479 (87.7)	483 (89.1)	
Yes	27 (4.9)	30 (5.5)	24 (4.4)	
Unknown	36 (6.6)	37 (6.8)	35 (6.5)	

UPF, ultra-processed food and drinks (based on group 4 of the NOVA classification); SD, standard deviation; IQR, interquartile range. ^1^ Categories based on sex-specific tertiles of the consumption of ultra-processed food and drinks. ^2^ Calculated as daily g of UPF/total daily g, multiplied by 100. ^3^
*p*-value for heterogeneity calculated with the ANOVA for continuous variables and with the chi-squared test for categorical variables. Missing values were excluded from these tests. ^4^ There were 68 controls with missing information on body mass index. ^5^ Physical activity in the last 10 years up to 2 years before diagnosis measured in METs/week: inactive (0), low (0.1–8), moderate (8–15.9) and very active (≥16). In bold: *p*-value < 0.05.

**Table 3 ijerph-18-05457-t003:** Association between the consumption of ultra-processed food (NOVA classification, group 4) and chronic lymphocytic leukemia overall by Rai stage and the time from diagnosis to interview in the MCC-Spain study.

			Consumption of Ultra-Processed Food (G4) ^1^					
	N Caes/Controls		Low OR (95% CI)	Medium OR (95% CI)	High OR (95% CI)	*p* for Trend	*10% Increase* OR (95% CI)	*p-Het* ^2^
**Total**								-
	320/1634	Model 1	1	1.02 (0.72; 1.45)	1.12 (0.79; 1.59)	0.54	1.09 (0.96; 1.24)	
	222/1566	Model 2	1	1.00 (0.69; 1.45)	1.09 (0.74; 1.60)	0.67	1.09 (0.94; 1.25)	
**Rai stage at diagnosis** ^3^								0.31
0	138/1634	Model 1	1	1.00 (0.65; 1.55)	1.02 (0.66; 1.60)	0.92	1.04 (0.88; 1.24)	
	131/1566	Model 2	1	1.03 (0.65; 1.63)	1.05 (0.65; 1.71)	0.83	1.05 (0.87; 1.25)	
1–4	88/1634	Model 1	1	1.14 (0.65; 1.99)	1.42 (0.83; 2.46)	0.20	1.19 (0.99; 1.43)	
	87/1566	Model 2	1	1.02 (0.57; 1.81)	1.24/0.69; 2.22)	0.46	1.16 (0.95; 1.42)	
**Time from diagnosis to interview**								0.08
<1 year (incident cases)	97/1634	Model 1	1	1.30 (0.77; 2.20)	1.46 (0.86; 2.47)	0.16	**1.23 (1.03; 1.46)**	
	94/1566	Model 2	1	1.38 (0.79; 2.41)	1.51 (0.85; 2.69)	0.17	**1.22 (1.02; 1.47)**	
Between 1 and 3 years (prevalent cases)	133/1634	Model 1	1	0.87 (0.56; 1.36)	0.92 (0.58; 1.44)	0.70	0.97 (0.81; 1.17)	
	128/1566	Model 2	1	0.79 (0.50; 1.27)	0.84 (0.52; 1.37)	0.49	0.95 (0.78; 1.16)	

N, number; OR, odds ratio; CI, confidence interval; P-het: *p*-values for heterogeneity.^1^ Categories based on sex-specific tertiles of the consumption of ultra-processed food and drinks. ^2^ Test of heterogeneity based on case-case analyses, assessed using the continuous UPF variable and the model 2 level of adjustment. ^3^ In four CLL cases, the RAI stage was not available hence they were excluded from the analyses by RAI stage. Model 1: logistic regression model adjusted for age, sex, province and educational level. Model 2: model 1 further adjusted for family history of hematological neoplasms, ever worked in farming, physical activity, energy intake, ethanol intake and smoking status. In bold: *p*-value < 0.05.

## Data Availability

Data are available on reasonable request. All data relevant to the study are included in the article or uploaded as online Appendix A.

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
