# Peer review of "Consumption of Ultra-Processed Food and Drinks and Chronic Lymphocytic Leukemia in the MCC-Spain Study"

_ijerph, 2021, doi:10.3390/ijerph18105457_

Round 1

Reviewer 1 Report

The study utilizes the food frequency questionnaire and appropriate statistic models to evaluate the association between the UPF intake and CLL. The study design is clear and straightforward. The manuscript is very well written and easy to understand. 

Specific comments:

(1) What is the aim for adding certain nutritional variables to the analysis as shown in Figure 1? It is not very clear what question the forest plot (Figure 1) planned to address. Besides, it is preferred to show what the left and right sides to the dashed line suggest/favor in the forest plot (Figure 1).

(2) How do you interpret the heterogeneity in Table 3, moderate, substantial or considerable based on the observed p-value? I2 statistics could be helpful.

(3) In Table 3, there is no OR value for Model 2 in "Rai stage = 0" and "Time from diagnosis < 1 year" under low UPF consumption group. How is the OR of Model 2 in medium and high UPF consumption groups calculated in these two lanes?

(4) I am wondering if other pre-existing health conditions like diabetes, cardiopathy etc also need to be considered in future studies.

Author Response

The study utilizes the food frequency questionnaire and appropriate statistic models to evaluate the association between the UPF intake and CLL. The study design is clear and straightforward. The manuscript is very well written and easy to understand. 

Specific comments:

(1) What is the aim for adding certain nutritional variables to the analysis as shown in Figure 1? It is not very clear what question the forest plot (Figure 1) planned to address. Besides, it is preferred to show what the left and right sides to the dashed line suggest/favor in the forest plot (Figure 1).

Following previous studies on UPF and cancer (Fiolet et al. 2018 and Romaguera et al. 2020), we assessed whether the associations found for UPF remained when dietary variables that could act as potential confounders were added. The idea is that is such inclusion did not attenuate the association, it might indicate that the association may be driven by factors beyond the dietary quality of such foods, such as food additives. This information is provided in the discussion section:

Line 306: “Several hypotheses have been suggested to explain the association between UPF consumption and cancer risk. One relates to the poorer nutritional quality of diets rich in processed foods, which tend to be rich in sugars, fats, salt, and excessive energy[2,33]. In our study, however, the association of UPF with CLL in incident cases remained statistically significant after adjustment for these dietary factors, suggesting, in line with other studies[8], the involvement of other components (e.g. additives or bioactive compounds derived from processing/packaging) beyond “nutritional quality”. Overall, mediation studies are needed to better understand the relative role of nutritional composition, additives, and neoformed contaminants in this relationship.”

We have modified the Figure 1 to better address the objectives of the forest plots. Here is the new version now included in the manuscript and the explanations:

SEE FIGURE 1 (in pdf) since it can't be included in this format.

Figure 1. Association between 10% increment in ultra-processed food and drink consumption and chronic lymphocytic leukemia in the MCC-Spain study, further adjusted for nutritional characteristics for all CLL cases (top graph) and restricted to cases recruited within 1 year of diagnosis (incident; bottom graph). OR, odds ratio; 95% CI, 95% confidence interval; CLL, chronic lymphocytic leukemia; UPF, ultra-processed foods and drinks. 1Logistic regression adjusted for age, sex, province, educational level adjusted for family history of hematological neoplasms, ever worked on farming, physical activity, energy intake, ethanol intake, and smoking status. Black squares and horizontal lines indicate OR and 95% CI, respectively.

(2) How do you interpret the heterogeneity in Table 3, moderate, substantial or considerable based on the observed p-value? I2 statistics could be helpful.

To assess whether results stratified according to Rai stage and time from diagnosis to interview where homogeneous, a case-case analysis was performed (we have added a footnote in Table 3 in order to clarify it).

For Rai stages, our data provide no evidence of heterogeneity between the severity groups (p-value for heterogeneity= 0.31). In relation to the second p-value for heterogeneity (p-value= 0.08) of Table 3, while we indeed used the conventional arbitrary cut-off of 5% to define statistical significance, we think that this result should still be emphasized. In the context of a study design including prevalent cases, a p-value of 0.08 should be taken into account. We would definitely not interpret it as moderate but it gives an interesting signal for future study in meeting strict design requirement (Matthews, 2021). We would tend to use I2 for meta-analysis comparing results from different studies.

Matthews, R. (2021), The p‐value statement, five years on. Significance, 18: 16-19. https://doi.org/10.1111/1740-9713.01505

(3) In Table 3, there is no OR value for Model 2 in "Rai stage = 0" and "Time from diagnosis < 1 year" under low UPF consumption group. How is the OR of Model 2 in medium and high UPF consumption groups calculated in these two lanes?

Thank you for the comment. It was a typographic error when preparing Table 3, we have now added the reference category “1”.

(4) I am wondering if other pre-existing health conditions like diabetes, cardiopathy etc also need to be considered in future studies.

We thank the reviewer for this comment. Chronic lymphocytic leukemia has been associated with autoimmune diseases and allergy (Slager SL et al. 2014). Since these conditions could affect dietary patterns, it might indeed be good advice to include them in future studies.  Unfortunately, the data on allergy and immune diseases were incomplete in our study and thus, could not be considered in these analyses. We have added this to the discussion:

Line 342: “Finally, although we adjusted for a range of potential confounders, residual confounding cannot be totally ruled out. In particular, we lacked data of several conditions, such as autoimmune diseases and allergy, that have been consistently associated with CLL[3], and that should ideally be considered in future studies.”

Reviewer 2 Report

This is a nice paper.

    - Please explain to the reader why the proportion UPF was of a person's diet was chosen as exposure, rather than the absolute amount of UPF. Similarly for the variables "sugar intake" and "saturated fatty acide intake"

    - Please explain to the reader the rationale for adjusting for mediators in the complementary analysis.

    - I felt that Table 2 was rather one-sided. There is no equivalant for the cases, which I think there should be. If you feel that you cannot add another table in the main body of the article, please consider moving table 2 to supplementary and adding an equivalent table for cases.

    - In the footnote to table 2 please make the reference numbers 1, 2 and 3 clearer - perhaps by making them bold and/or place them in superscript position.

    - It might be worth mentioning that those who have had a CLL diganosis for longer than one year, and who have changed their diet after diagnosis, would report their changed diet. This might explain the discrepancy between your findings for incident cases, and those of a longer standing diagnosis.

    - Table S2. What is "P-int"?

Author Response

This is a nice paper.

    - Please explain to the reader why the proportion UPF was of a person's diet was chosen as exposure, rather than the absolute amount of UPF. Similarly for the variables "sugar intake" and "saturated fatty acide intake"

Most studies that have previously assessed the association between UPF consumption and different health outcomes, and particularly those focused on cancer (Fiolet et al. 2018 and Romaguera et al. 2020), performed their analyses using the proportion of UPF in each individual’s diet. We followed the same methodology in order to ease comparability across studies. We now indicate such information in line 166.

Line 166: “Following previous studies [24,25], for each individual, consumption of all items included in each NOVA group (expressed in daily grams) was added and expressed as a percentage of total dietary intake (daily g within NOVA category/total daily g, multiplied by 100).”

    - Please explain to the reader the rationale for adjusting for mediators in the complementary analysis.

Following previous studies on UPF and cancer (Fiolet et al. 2018 and Romaguera et al. 2020), we assessed whether the associations found for UPF remained when dietary variables that could act as potential confounders were added. The idea is that if such inclusion did not attenuate the association, it might indicate that the association may be driven by factors beyond the dietary quality of such foods, such as food additives. This information is now provided in the discussion section:

Line 306: “Several hypotheses have been suggested to explain the association between UPF consumption and cancer risk. One relates to the poorer nutritional quality of diets rich in processed foods, which tend to be rich in sugars, fats, salt, and excessive energy[2,33]. In our study, however, the association of UPF with CLL in incident cases remained statistically significant after adjustment for these dietary factors, suggesting, in line with other studies[8], the involvement of other components (e.g. additives or bioactive compounds derived from processing/packaging) beyond “nutritional quality”. Overall, mediation studies are needed to better understand the relative role of nutritional composition, additives, and neoformed contaminants in this relationship”.

    - I felt that Table 2 was rather one-sided. There is no equivalant for the cases, which I think there should be. If you feel that you cannot add another table in the main body of the article, please consider moving table 2 to supplementary and adding an equivalent table for cases.

Table 2 is based only on controls to examine the existence of potential confounder variables. Following the recommendation of the reviewer, we now provide the same table restricted to cases in supplementary material, Table S2.

Supplementary material, Table S2:

Table S2: Characteristics of cases according to consumption of ultra-processed foods (NOVA classification, group 4) in the MCC-Spain study.

Ultra-processed food and drinks consumption (G4)1

Only cases (n=230)

Low

(n=76)

Medium

(n=75)

High

(n=79)

P-value2

Proportion of UPF consumption [(g/ g total intake) *100], mean (SD)

5.0 (1.8)

11.2 (2.5)

26.6 (11.8)

<0.001

Province, n(%)

0.303

Barcelona

44 (57.9)  

51 (68.0)   

53 (67.1)  

Asturias

16 (21.1)  

6 (8.0)    

7 (8.9)   

Cantabria

2 (2.6)   

3 (4.0)    

2 (2.5)   

Granada

8 (10.5)   

8 (10.7)   

6 (7.6)   

Girona

6 (7.9)   

7 (9.3)   

11 (13.9)  

Age (years), mean (SD)

67.9 (8.3)

66.3 (10.7)

62.9 (10.6)

0.007

Sex, n(%)

0.521

Male

48 (63.2)

52 (69.3)

48 (60.8)  

Female

28 (36.8)

23 (30.7)

31 (39.2)

Education, n(%)

0.010

Primary

55 (72.4)

37 (49.3)

37 (46.8)

Secondary

14 (18.4)

21 (28.0)   

27 (34.2)

University

7 (9.2)   

17 (22.7)  

15 (19.0)   

Body mass index3 (kg/m2), mean (SD)

26.7 (3.9)  

27.2 (4.3)

28.0 (4.3)

0.136

Energy intake (kcal/day), mean (SD)

1763.9 (524.9)

1953.2 (573.0)

2191.1 (667.3)

<0.001

Alcohol consumption, median (IQR)

3.5 (0.0-10.5)

6.0 (1.1-11.5)

5.1 (0.4-12.6)

0.991

Smoking status, n(%)

0.229

Never

27 (35.5)

29 (38.7)

37 (46.8)

Former

10 (13.2)  

12 (16.0)   

16 (20.3)  

Current

39 (51.3)  

34 (45.3)  

25 (31.6)  

Unknown

1 (1.3)

Physical activity4, n(%)

0.972

Inactive

30 (39.5)  

28 (37.3)

29 (36.7)

Moderately active

10 (13.2)  

10 (13.3)

12 (15.2)  

Active

9 (11.8)   

8 (10.7)   

12 (15.2)  

Very active

27 (35.5)  

28 (37.3)  

25 (31.6)  

Unknown

1 (1.3)   

1 (1.3)   

Ever worked in farming or agriculture, n(%)

0.011

No

40 (52.6)  

50 (66.7)  

61 (77.2)

Yes

36 (47.4)  

25 (33.3)  

17 (21.5)

Unknown

1 (1.3)   

Family history of hematological malignancy, n(%)

0.652

No

65 (85.5)  

63 (84)   

65 (82.3)  

Yes

6 (7.9)   

10 (13.3)

10 (12.7)

Unknown

5 (6.6)   

2 (2.7)   

4 (5.1)   

    - In the footnote to table 2 please make the reference numbers 1, 2 and 3 clearer - perhaps by making them bold and/or place them in superscript position.

It has been modified in order to make it clearer.

    - It might be worth mentioning that those who have had a CLL diganosis for longer than one year, and who have changed their diet after diagnosis, would report their changed diet. This might explain the discrepancy between your findings for incident cases, and those of a longer standing diagnosis.

Unfortunately, we lacked complete and detailed data of dietary changes that took place during the previous years before the interview. Therefore, we were unable to further examine the impact of such changes as a potential explanation of the discrepancies observed between incident and prevalent cases.

We have added this point in the abstract section:

Line 57:  “…suggesting that overall results might be affected by the inclusion of prevalent cases, which might have changed their dietary habits after cancer diagnosis.”

And discussion:

Line 300: “Although CLL is typically an indolent disease, we cannot totally rule out that the inclusion of prevalent cases might have attenuated our estimates due to survival bias or, furthermore, dietary changes after diagnosis. Thus, results in incident cases point to a potential association between consumption of UPF and CLL which should be clarified in further studies, ideally with a prospective design and more statistically power.”

Line 338: “This is particularly relevant for prevalent cases, in case that they changed their habitual diet after cancer diagnosis.”

    - Table S2. What is "P-int"?

It is the p-value for interaction, which was calculated by modelling cross-product terms between UPF consumption (as continuous variable) and sex, age, smoking status, and alcohol intake. We have added a footnote in Table S2 in order to clarify it.

Reviewer 3 Report

Reviewers Comments to Authors

Paper number: ijerph-1201660

Paper title: Ultra-processed food and drinks consumption and chronic lymphocytic leukemia in the MCC-Spain study

Type and Category: Article

  1. Discussion of specific areas for improvement
  • Major Issues
    1. The paper title and contents could be reconsidered the relevance.
    2. There is lack of theoretical research background and research questions in introduction, also lacking of clear purposes and description of methodology and findings as a map to guide the audiences.
    3. The abstract can be re-organized to draw an outline and clarify the framework of this manuscript.
    4. The conclusions should cope with abstract and introduction, the comment to author is to re-organize and restructure the abstract and introduction.
    5. This paper has well typography, spelling, grammar, and phrasing issues.
    6. The manuscript can add more insight theoretical and practical applications.

Author Response

Reviewers Comments to Authors

Paper number: ijerph-1201660

Paper title: Ultra-processed food and drinks consumption and chronic lymphocytic leukemia in the MCC-Spain study

Type and Category: Article

  1. Discussion of specific areas for improvement
  • Major Issues
    1. The paper title and contents could be reconsidered the relevance.
    2. There is lack of theoretical research background and research questions in introduction, also lacking of clear purposes and description of methodology and findings as a map to guide the audiences.
    3. The abstract can be re-organized to draw an outline and clarify the framework of this manuscript.
    4. The conclusions should cope with abstract and introduction, the comment to author is to re-organize and restructure the abstract and introduction.

We thank the reviewer for their comments. Since the issues 1 to 4 can be joined into structural and descriptive issues, we will provide a joint answer about them. 

In relation with the title, we have slightly modified it in this way “Consumption of ultra-processed food and drinks and chronic lymphocytic leukemia in the MCC-Spain study”. We would like to keep this title as this work goes in parallel with other work that was done within the MCC-Spain study on breast and prostate cancer (Romaguera et al, 2021). 

We have further developed the relevance of this work and the structure of the manuscript. In particular:

  • in the abstract:

“Chronic lymphocytic leukemia (CLL) is the most common leukemia in adults in Western countries. It etiology is largely unknown, but increasing incidence rates observed worldwide suggest that lifestyle and environmental factors, such as diet, might play a role in the development of CLL. Hence, we hypothesized that consumption of ultraprocessed food and drinks (UPF) might be associated with CLL. Data from a Spanish population-based case-control study (MCC-Spain study), including 230 CLL cases (recruited within three years of diagnosis) and 1634 population-based controls were used. Usual diet during the previous year was collected through a validated food frequency questionnaire, and food and drinks consumption was categorized using the NOVA classification scheme. Logistic regression models adjusted for potential confounders were used. Overall, no association was reported between UPF consumption and CLL cases (OR per each 10% increase of UPF relative contribution to total dietary intake = 1.09 (95% CI: 0.94; 1.25)), independently of Rai stage at diagnosis. However, when analyses were restricted to cases diagnosed within <1 year (incident), each 10% increment in UPF consumption was associated with a 22% higher odds of CLL (95% CI: 1.02, 1.47), suggesting that overall results might be affected by the inclusion of prevalent cases, which might have changed their dietary habits after cancer diagnosis. Given the low number of cases in subgroup analyses and multiple tests performed, chance findings cannot totally be ruled out. Nonetheless, positive associations found in CLL incident cases merit further research, ideally in well-powered studies with a prospective design.”

  • and in the introduction (lines 66-88, and 98-110):

“Chronic lymphocytic leukemia (CLL), a subtype of non-Hodgkin lymphoma (NHL), is the most common leukemia in adults in Western countries, with an incidence rate of 5 per 100,000 person-years in Europe[1]. The disease, characterized by an abnormal accumulation of B-lymphocytes in blood, bone marrow and/or lymphoid tissues, typically occurs in the elderly and shows a highly heterogeneous clinical course. While the majority of patients live for decades without being treated and die from other causes, others show a rapidly aggressive course and premature death [2]. In line with other hematological neoplasms, the etiology of CLL is largely unknown, with few well-established risk factors including increasing age, male sex, being Caucasian, and a familial history of hematological neoplasms, and several suggestive associations yet to be confirmed[3]. However, the high incidence of CLL in Western regions[4], the increasing CLL incidence rates across the world [5], and the changing in CLL incidence patterns with acculturation as seen in higher incidence rates among Asian US born than Asian foreign born [6,7] suggest that lifestyle and environmental factors may also play an important role CLL etiology.

As regards dietary exposures, the 2007 report by the World Cancer Research Fund/American Institute for Cancer Research (WCRF/AICR)[8], pointed out a suggestive inverse association between NHL and consumption of vegetables, fruit, and alcoholic beverages, and a positive association with meat, total fat, body fatness, and dairy. Recent meta-analyses further support these associations [9-13] but data is still inconclusive, particularly for specific NHL subtypes. Specifically in CLL, evidence regarding dietary exposures and risk of CLL, which mainly arises from studies assessing single-food items[14], is inconsistent. […]”

“[…] However, other less desirable properties[17,18], including their poor nutritional composition (i.e. high amounts of saturated fats, refined starches, free sugar, and salts), the potential presence of components derived from food processing (i.e. heterocycle amines, acrylamide, or aromatic polycyclic hydrocarbons) or packaging (i.e. bisphenol A), might have a deleterious effect on human health[17]. Indeed, cumulative evidence from epidemiological studies points to an association between consumption of UPF and obesity[19], cardiovascular disease[20], mortality[21-23] and, more recently, cancer[24,25]. Regarding the latter, a French prospective study conducted in 104,980 participants, reported an increased risk of developing cancer and, more specifically, breast cancer [24]. Lately, in a large Spanish multicase-control study (MCC-Spain), including cases of colorectal (n=1,852), breast (n=1,486), and prostate cancer (n=953), consumption of UPF was associated with colorectal cancer [25]. However, to the date, no study has assessed its potential impact on hematological neoplasms.”

Finally, to better align the conclusion with the abstract content we have modified the conclusion such as:

“Findings from this Spanish population-based case-control study do not conclusively support an association between UPF consumption and CLL, yet results may be biased due to the inclusion of prevalent cases. The association reported in analyses restricted to incident cases points to a potential role of UPF in CLL. Further studies, ideally with a prospective design and more statistical power, are required to confirm these findings.”

    1. This paper has well typography, spelling, grammar, and phrasing issues.

We have checked the manuscript for English issues.

    1. The manuscript can add more insight theoretical and practical applications.

We are unsure what the reviewer is asking in this point. Could the reviewer please clarify this?
